# Employment Barriers for Racialized Immigrants: A Review of Economic and Social Integration Support and Gaps in Edmonton, Alberta

Doriane Intungane [1,*], Jennifer Long [1], Hellen Gateri [2] and Rita Dhungel [3]

1   Department of Anthropology, Economics, and Political Science, Faculty of Arts & Science, MacEwan University, Edmonton, AB T5J 4S2, Canada; longj34@macewan.ca
2   School of Social Work, MacEwan University, Edmonton, AB T5J 4S2, Canada; gaterih@macewan.ca
3   School of Social Work and Human Services, University of the Fraser Valley, Abbotsford, BC V2S 7M8, Canada; rita.dhungel@ufv.ca
*   Correspondence: intunganed@macewan.ca

**Abstract:** This article explores the strategies used by government-sponsored institutions dedicated to addressing systemic barriers to employment for racialized immigrants in Edmonton. The research involved conducting in-depth semi-structured interviews with service providers, employment program coordinators from different settlement and employment agencies, and a research and training centre operating in Edmonton, Alberta. The first objective is to understand the barriers racialized immigrants face through the hiring and promotion process. The second objective is to understand the support provided by those institutions and the impact of their equity policies on how they assist racialized Canadians in finding gainful employment. Lastly, this study explores the impact of the COVID-19 pandemic and the Black Lives Matter movement on the employment of racialized immigrants in Edmonton. The results show that around 50% of employment service providers acknowledged that visible minority immigrants face barriers while integrating into the labour market, including racial microaggressions in their jobs. In addition, the findings indicate a lack of programs tailored to the needs of racialized job seekers. Participants in this study reported that the Black Lives Matter movement raised awareness among employers regarding racial issues in the workplace. Hence, there is a demonstrated need for employers to undergo training to recognize and address racism in hiring, promoting, and retaining racialized employees at Canadian workplaces. Interviewees recognized that the COVID-19 pandemic negatively impacted racialized employees and newcomers. They recommended that Canadian companies establish educational programs that emphasize the importance and benefits of racial diversity, equity, and inclusion in the hiring process.

**Keywords:** barriers to labour-market integration; diversity and inclusion in Canadian companies; Edmonton; Alberta

## 1. Introduction

The Canadian federal government supports an immigration policy to increase the number of permanent residents as a strategy to combat Canada's ageing workforce (Centre for Race and Culture 2021). Canadian immigration admission criteria favour highly educated immigrants who are expected to "positively contribute to the economy and enhance global competitiveness" (Wilkinson et al. 2016, p. 6). The federal government has devised several programs to bring in economic-class immigrants, such as (1) the Canadian Experience Class, (2) Federal Skilled Worker Program, and (3) Federal Skilled Trades Program. These three programs traditionally represent the largest annual immigration category, with 164,416 economic immigration admissions in 2021 (Government of Canada 2022).

While the federal government is the policy engine behind much of Canada's immigration, recent federal policies have endeavoured to disseminate immigration strategies

through such schemes as the 2021 Municipal Nominee Program (MNP). This program seeks to "allow local communities, chambers of commerce, and local labour councils to directly sponsor permanent immigrants" (Government of Canada 2022). However, for Canadians to benefit from the MNP, we need to provide immigrants with job opportunities and a welcoming environment to see themselves as part of the local community.

Recent immigration trends to Canada showcase not only a concentration of newcomers in census metropolitan areas but also the growing ethnocultural diversity of newcomers (Zhuang 2021). Hence, municipalities are responsible for providing settlement services and the (social and physical) infrastructure to integrate newcomers (Zhuang 2021). They also need to address the (continued) existence of prejudice and discrimination toward immigrants, which affects the economic, sociocultural, and civic–political future of newcomers to Canada, as well as the perception of Canada as a receiving society or a society of settlement (Esses 2021). Discrimination is a significant issue to address, considering its impact on the Canadian economy. Reitz et al. (2014) found that the un- and underemployment of immigrants cost the Canadian economy CAD $11.37 billion annually in comparison to equally qualified non-immigrants (Esses 2021, p. 517).

Despite ameliorating initiatives in federal, provincial, and corporate settings that include regulations around credential assessment, professional licensing, and bridge training programs (Ng et al. 2006), racialized Canadians experience disproportionate negative evaluation of qualifications that could only be attributed to racial discrimination (Reitz et al. 2014). Visible minority refers to "persons, other than (Indigenous) peoples, who are non-Caucasian in race or non-white in colour" according to the Employment Equity Act (Statistics Canada 2023).[1] In 2017, Statistics Canada reported that visible minorities of working age will make up between 34.7% and 39.9% of the population in 2036 (Statistics Canada 2017a). Studies have demonstrated the existence of labour market discrimination toward visible minorities in Canada; examples include the gap in income earnings across generation of immigrants (Skuterud 2010); discrepancies in employment rates (Block et al. 2014; Lightman and Gingrich 2018); and employment and role segregation (Guo 2015; Wilkinson et al. 2016). In fact, Statistics Canada reported a higher unemployment rate of racialized workers, such as Arab and Black (17.9% and 17.6%, respectively) in the summer 2020 in comparison to the average unemployment rate (11.1%) (Public Health Agency of Canada 2021). It is in this specific context that we seek to explore the strategies of employment service providers in promoting diversity and inclusion in the workplace. Canada relies on immigrants to maintain a high level of economic growth (Akbari and Haider 2018). Without effective change, discriminatory employment practices will have negative effects on the economic growth of the country, not to mention the well-being and quality of life of new Canadians.

Our research goals are threefold: (1) To provide an up-to-date understanding of employment of racialized individuals in Alberta in the 2000s; (2) To identify the impact of employment equity-related policies for racialized Canadians' and newcomers' employment, and (3) To better understand how racialized individuals searched and found gainful employment and the role of employment and settlement agencies in the job search process. With the onset of the pandemic, the project objectives shifted to discuss the impact of COVID-19 and the Black Lives Matter 2020 campaign on racialized Edmontonians' experiences of employment.

In the following discussion, we provide a brief background on the policies affecting immigrant diversity and economic integration at federal, provincial, and municipal levels. The section also includes a short overview of racialized immigrant admissions to the country using the most recent publicly available data through Statistics Canada. We then provide an overview of selected literature and review our methods and research questions. This work is followed by a summary of qualitative data gathered to supplement the paucity of material available regarding initiatives to support racialized immigrants' economic and social integration in Edmonton, Alberta.

## 2. Policies and Demographic Background

Although the existing research shows data disparities among different groups of immigrants and Canadians participating in the labour force, the federal government has adopted various policies and passed acts to increase diversity and ensure equal opportunity in the Canadian labour force. The *Canadian Multiculturalism Act* was adopted to help all individuals, including minority groups, preserve their cultural freedom and to recognize everyone's contribution to Canadian society [R.S.C., 1985, c. 24 (4th Supp.)]. Jeyapal (2018) argues that this policy aligned with the Liberal Party initiatives for citizenship and inclusion, which indoctrinated racialized others into a narrative of cultural practices and ceremonies that did not address the structural issues of racial equity or the eradication of racial discrimination. Thobani (2007) further argues that the success of multiculturalism lies in its facilitation of the integration of immigrants on the nation's terms without considering the struggles of people of colour against the racism of the nation-state. In 1977, the federal government passed the *Canadian Human Rights Act* to ensure that all individuals have an equal opportunity to make the lives they wish, consistent with their duties and obligations as members of society, without being hindered or prevented from doing so by discriminatory practices (c. 33, s. 1). Although the Act reiterates the discourse of equality across multiple social demographics, according to Mullings (2009) race-based discrimination continues to be reproduced by the Canadian Human Rights Tribunal. Mullings described how race, racialization, and racism are complicated through organisational processes and are reproduced by the tribunal through institutional practices to legitimise existing power relations. Mullings (2009) defines racism as the idea of how groups are described ethnically and the way that ethnic groups are viewed as inferior: "Racism enables certain groups to be formed and valued as undesirable, which, in turn, causes these groups to be assimilated, excluded and exterminated", while racialization is the process of the exclusion and unequal treatment of different groups based on the racism ideology (Human Rights and Equity Office n.d., p. 12). The *Employment Equity Act* exists to promote equality of opportunities and prevent employment discrimination based on race, ethnic origin, religion, age, sexual orientation, and gender identity (Government of Canada 1995). However, the Act lacks formal accountability and consequences of systemic discrimination practices by employers, which places the onus on racialized people to demonstrate the existence of racism through an adversarial process (Jeyapal 2018). This reality highlights the structural disconnect of policies that claim equity and continues to operate through race-neutral logics that disadvantage racialized people facing employment discrimination. The federal government published *Building a Foundation for Change: Canada's Anti-Racism Strategy 2019–2022* (Canadian Heritage 2019) to empower communities, build awareness of historical racism and contemporary racist practices, and change attitudes. As part of this strategy, the federal government committed $4.6 million dollars to build an Anti-Racism Secretariat to "engage with Indigenous Peoples and partners to identify and develop further areas for action" (p. 16). Specific to employment, this strategy sought to reduce "barriers to hiring, leadership training, and workplace skills training, including encouraging partnerships between employers and employees in reducing barriers" (p. 12).

Although the existing research shows data disparities among different groups of immigrants and Canadians participating in the labour force, the federal government has adopted various policies and passed acts to increase diversity and ensure equal opportunity in the Canadian labour force. The *Canadian Multiculturalism Act* was adopted to help all individuals, including minority groups, preserve their cultural freedom and to recognize everyone's contribution to Canadian society [R.S.C., 1985, c. 24 (4th Supp.)]. Jeyapal (2018) argues that this policy aligned with the Liberal Party initiatives for citizenship and inclusion, which indoctrinated racialized others into a narrative of cultural practices and ceremonies that did not address the structural issues of racial equity or the eradication of racial discrimination. Thobani (2007) further argues that the success of multiculturalism lies in its facilitation of the integration of immigrants on the nation's terms without considering the struggles of people of colour against the racism of the nation-state. In 1977, the federal govern-

ment passed the *Canadian Human Rights Act* to ensure that all individuals have an equal opportunity to make the lives they wish, consistent with their duties and obligations as members of society, without being hindered or prevented from doing so by discriminatory practices (c. 33, s. 1). Although the Act reiterates the discourse of equality across multiple social demographics, according to Mullings (2009) race-based discrimination continues to be reproduced by the Canadian Human Rights Tribunal. Mullings described how race, racialization, and racism are complicated through organisational processes and are reproduced by the tribunal through institutional practices to legitimise existing power relations. Mullings (2009) defines racism as the idea of how groups are described ethnically and the way that ethnic groups are inferiorized: "Racism enables certain groups to be formed and valued as undesirable, which, in turn, causes these groups to be assimilated, excluded and exterminated", while racialization is the process of the exclusion and unequal treatment of different groups based on the racism ideology (Human Rights and Equity Office n.d., p. 12). The *Employment Equity Act* exists to promote equality of opportunities and prevent employment discrimination based on race, ethnic origin, religion, age, sexual orientation, and gender identity (Government of Canada 1995). However, the Act lacks formal accountability and consequences of systemic discrimination practices by employers, which places the onus on racialized people to demonstrate the existence of racism through an adversarial process (Jeyapal 2018). This reality highlights the structural disconnect of policies that claim equity and continues to operate through race-neutral logics that disadvantage racialized people facing employment discrimination. The federal government published *Building a Foundation for Change: Canada's Anti-Racism Strategy 2019–2022* (Canadian Heritage 2019) to empower communities, build awareness of historical racism and contemporary racist practices, and change attitudes. As part of this strategy, the federal government committed $4.6 million dollars to build an Anti-Racism Secretariat to "engage with Indigenous Peoples and partners to identify and develop further areas for action" (p. 16). Specific to employment, this strategy sought to reduce "barriers to hiring, leadership training, and workplace skills training, including encouraging partnerships between employers and employees in reducing barriers" (p. 12).

Provincially, Alberta has followed suit and created legislation to combat discrimination in the workplace; this included the *Individual Rights Protection Act,* which led to the creation of the provincial *Human Rights Commission* in 1972. Over two decades later, the province created the *Alberta Human Rights Act* (then called the *Human Rights, Citizenship, and Multiculturalism Act*) that prohibited "discrimination in employment based on the protected grounds" (Alberta King's Printer 2023). In 2018, the New Democratic Party (NDP) provincial government developed the Anti-Racism Advisory Council following the outcome of the *Taking Action Against Racism Plan*. The council's goal was to advise the Minister of Education on combating racism through anti-racism advocacy (Smith and Raphael n.d., p. 15). Under United Conservative Party leadership, however, the provincial strategy to anti-racism has been met with criticism. Described as "weak" and lacking meaningful engagement, the recently released (2022) strategy rejected calls for race-based data collection across government departments and police services, and (according to Alberta NDP multiculturalism critic) lacked a comprehensive action plan (Johnson 2022).

Municipally, there are very few policies that specifically address gaps in racialized employment. However, in 2007, the City of Edmonton adopted City Policy C529 that clearly stated the city's commitment to building a municipal environment that "attracts and retains immigrants, refugees and their families" (City of Edmonton 2007, p. 1). More recently, in early 2020, the City of Edmonton has followed the lead of the province (and all other municipalities across Canada) and convened a municipal Anti-Racism Advisory Committee. The goal of this committee is to "raise awareness and catalyse action on racism and anti-racism in Edmonton and provide advice to [City] Council regarding community perspectives on issues relating to racism" (City of Edmonton 2020, para. 1). While the focus of this body might incorporate equitable employment, it is not an overtly stated goal or priority area.

It is worth noting that Canada funds various settlement programs and institutions to aid the community integration process for immigrants. For example, Canada funds different institutions to offer training in language and culture to refugees and permanent residents; for example, Language Instruction for Newcomers to Canada (LINC) programs provide English and French lessons across Canada (Government of Canada 2018; Myers and Conte 2013). There are also multiple settlement service agencies offering programs to help immigrants integrate more easily into the labour force. Despite the existence of many programs and institutions supporting immigrants in Canada, it is still unclear how effective they are in helping job seekers who are experiencing racial discrimination.

It might seem as though there are enough policies and strategies to help visible minority immigrants find employment; however, studies demonstrate that job seekers with ethnic sounding names, non-Canadian English accents, and foreign education have fewer employment opportunities and are more likely to be un/underemployed and earn less compared to their non-visible minorities peers (refer to Oreopoulos 2011; Pendakur and Pendakur 2011; Premji et al. 2014; Branker 2017; Banerjee et al. 2018). Figure 1 uses census data to compare the unemployment rate of visible and non-visible immigrants in Canada and in the four provinces who welcome the most immigrants (British Columbia [BC], Ontario, Quebec, and Alberta). According to the Government of Alberta website (2023), 40.3% of Edmonton's general population identified as a visible minority, which was identified as the 4th highest in the province (para. 1). The authors note that "the percentage of the population identifying as a visible minority in Edmonton greatly increased 10.8% in the last five years" (Government of Alberta 2023, para. 1). (See also (Bilodeau et al. 2015)).

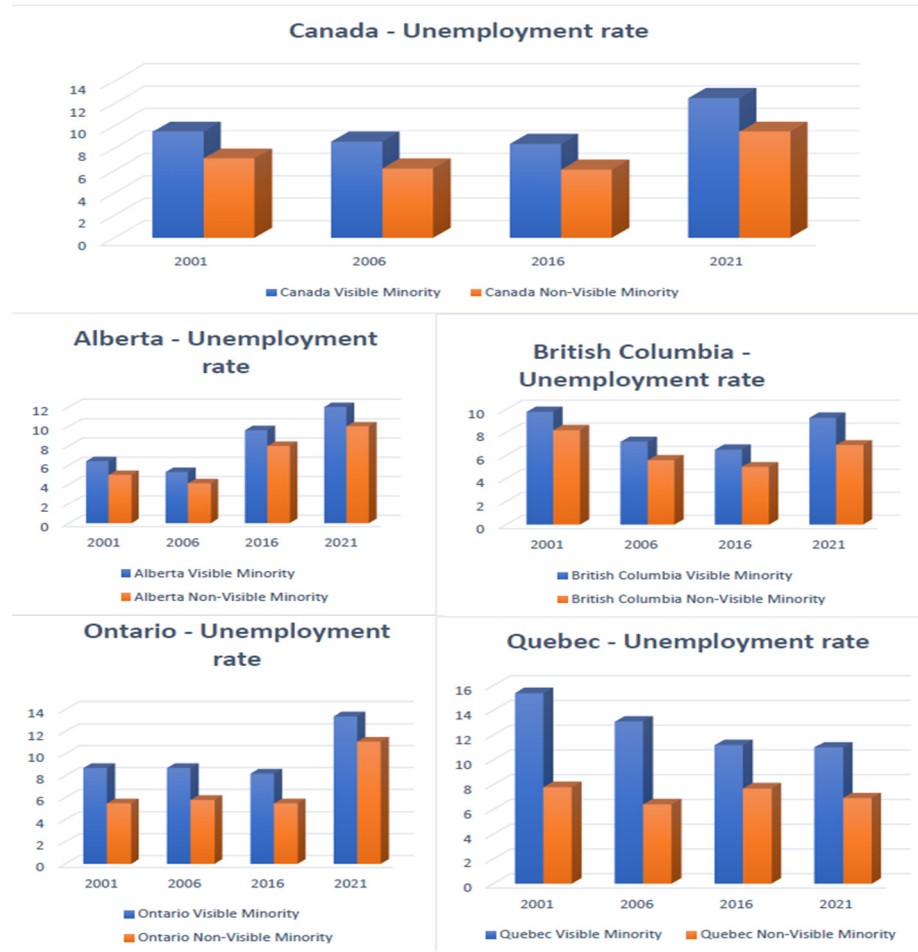

**Figure 1.** Unemployment rate of visible- Visible and non-visible minorities [Census 2001 (Statistics Canada 2003), 2006 (Statistics Canada 2008), 2016 (Statistics Canada 2017b), 2021 (Statistics Canada 2022)].

The unemployment rate for visible minority immigrants is higher than the unemployment rate for non-visible minority immigrants in all provinces. From 2001 to 2016, Quebec had the highest unemployment rate for visible minority immigrants. Before the 2016 census, Alberta had the lowest unemployment rate for visible minority immigrants (compared to other provinces); however, differences in unemployment rates can be explained by economic cycle differences across provinces. Over the years, all provinces except for Alberta have reduced the gap in the unemployment rate between visible and non-visible minority immigrants over the years. Quebec has had a remarkable decrease in this rate of around 53%. Figure 1 shows also that the unemployment rate of visible minority immigrants was still high in all provinces in 2021.

Racial discrimination is also found in senior management occupations data and income of visible and non-visible minority workers across Canada. The national occupation classification showed that the rate of racialized immigrants in senior managerial positions is lower than the rate of other immigrants. In the 2016 census, around 11.5% of visible minorities occupied the senior manager at the country level. Less than 10% of visible minority immigrants are senior managers in Alberta, while the rate is 21.5%, 15.5%, and 5.6% in BC, Ontario, Quebec, respectively (see the Employment and Social Development Canada 2016). Cardozo and Pendakur (2008) and Javdani (2020) have demonstrated that racialized immigrants have a lower probability of being promoted than other immigrants. The management occupation has also been shown to be lower for visible minorities compared to other workers. The 2021 census shows the existence of inequality based on income as well. In the four provinces considered above and at the country level, there is an income gap between visible and non-visible immigrants (Statistics Canada 2022).

## 3. Literature Review

There is extensive literature analysing the labour market experience of racialized immigrants in Canada and around the world. The goal of this paper is not to enumerate all the studies which investigate the employment conditions of racialized immigrants. Instead, we will summarise the main issues of racialized immigrants' experience in the labour market in Canada and review the state of the literature on racialized exclusion specifically in Edmonton, Alberta.

Ogilvie et al. (2007), Nadeau and Seckin (2010), Grenier and Nadeau (2011), and the World Education Services (2019) identify lack of language proficiency, lack of Canadian work experience, a systematic unwillingness to recognize certain foreign credentials, and low income as some of the most significant issues that immigrants face in the labour market. Scholars have argued that racialized immigrants face particular intersectional barriers related to the country of origin, racial identity, and religious difference; these factors play an important role in the exclusion of newcomers and Canadian immigrants from the Canadian labour market (refer to Esses et al. 2006; Thomas 2021). Boyd and Thomas (2002) explain that minority immigrants struggle more in the labour market because their education and credentials are undervalued compared to the education of non-visible minority immigrants (see also Nangia and Arora 2021). The literature points to the origins of immigrants and country of education as key indicators to their successful integration into the labour market (Gilmore and Le Petit 2008). Esses et al. (2006), for example, conclude that the lack of a clear foreign credential assessment allows prejudice to affect the evaluation of qualifications and skills of visible minorities. Oreopoulos (2011) demonstrates that having foreign education and a foreign-sounding name is more likely to reduce the number of call-backs from potential employers. Similarly, Zaami and Madibbo (2021) and Branker (2017) demonstrate that racialized immigrants are excluded during the hiring process when their names sound "non-white" or they speak English with an accent. Zaami and Madibbo's (2021) differentiate accents as "sounding Canadian" and "sounding Black" accents; their experiments are based on the job searches and job experiences of Black African youth in Calgary. Branker's (2017) research focuses on the experiences of Caribbean immigrants living in neighbouring cities of Toronto.

Another particularity of racialized immigrants in the labour market is related to earnings. Cardozo and Pendakur (2008) and Pendakur (2005) demonstrate a clear difference between visible minority earnings and non-visible minority earnings at the national and municipal levels in major cities welcoming more immigrants. According to Goldring and Joly (2014), income distribution varies based on birth, race, and legal status, amplifying the precarity of employment in southern Ontario. The authors found that racialized immigrants experience precarious work resulting in low and unstable income or wages (refer also to Nangia and Arora 2021). Khattab et al. (2019) argue that racialized individuals who are also visibly religious (such as Muslims who experience a "Muslim penalty" in the Canadian labour market), experience further exclusion above and beyond those racialized individuals who might otherwise be categorised as without religious affiliation. Reitz et al. (2022) have found similar penalties in earnings for first-generation racialized Muslim immigrants to Canada. This work highlights the intersecting biases of Islamophobia and racism and their impact on employment experiences.

On a municipal level, there is less data and fewer research studies available on racialized newcomers' integration into the labour market or the experiences of racialized Canadians in the workplace (Agrawal 2017). Guo (2015) argues that racialized exclusion in Edmonton (and Calgary) results from institutional barriers that cause a "triple threat" of un/underemployment, poor economic performance, and downward social mobility. Higginbottom (2000) also found that racialized nurses who were internationally trained reported negative experiences of "recruitment, reception, and support on arrival, stemming largely from their own unmet expectations" (as cited in Argawal 2017, p. 8).

In their *State of Immigration and Settlement in Edmonton Annual Report* (2021), the City of Edmonton identified their Economic Action Plan (enacted in April 2021) which sought to pay special attention to 'those left behind' in the context of post-COVID-19 recovery. The authors of this report advocated that the City of Edmonton pay particular attention to the inclusion of immigrants in their post-COVID-19 recovery work, and highlighted the need to address the implicit biases that negatively affected the hiring and promotion of racialized immigrants in Edmonton workplaces (City of Edmonton 2021, pp. 15, 22). This municipal report reveals that the City is aware of the impact of intersectional exclusions that result in the unemployment and underemployment of racialized newcomers and Edmontonians.

The above literature demonstrates the precarious employment environment facing racialized Canadians. This also begs the question as to how racialized Canadians experienced employment during the COVID-19 global pandemic. Although the literature on the impact of COVID-19 is not very extensive, various studies have already demonstrated the adverse social and economic effects of the pandemic on the health and employment of racialized Canadians. Kemei et al. (2023) found that Black Canadians experienced higher rates of infections, hospitalisation, and death in comparison to non-racialized Canadians. As racialized workers were among those (most) precariously employed in Canada before the pandemic, this group struggled to offset job loss or isolation in case of infection because they often live in smaller houses and share spaces with other family members. Mo et al. (2020) found that the COVID-19 pandemic had a differential socio-economic impact on historically marginalised groups. As in other studies, Mo et al. (2020) show that vulnerable individuals, including racialized workers and business owners, were negatively impacted by the pandemic. Hou et al. (2020) similarly found that low-paid workers' employment losses were significantly high for racialized Canadians.

The literature demonstrates the impact of longstanding systemic bias and exclusion against racialized individuals negatively affects their employment and health outcomes and how these biases deepened inequitable socioeconomic trends during the pandemic. Scholars explored how, if at all, the growth in awareness among majority community members around the (continued) existence of anti-Black racism and corporate responses to the Black Lives Matter campaigns in 2020 affected the employability of racialized individuals in Canada. Potvin (2020), Williams et al. (2019), and Houle (2020), for example, proved the existence of racism with some relation to poverty, precarious employment, and low quality

of living in Canada. Houle (2020) demonstrated the continuation of low-income status beyond the first generation into second- and third- generation Black children. In response to the increased attention paid to anti-Black racism, Smith and Rohde (2023, p. 6) developed a guide on behalf of the Canadian Human Rights Commission, describing how Canadian employers could "educate effectively, create relevant strategies and implement measures" that built inclusive and equitable workplaces that prevented and interrupted "any display of racist behaviours" (p. 6). Yet, there is little to no research on how effective such recommendations have been in addressing racism and bias in Canadian workplaces. This article seeks to contribute to the growing literature exploring this nexus of the longstanding systemic bias and racism in Canada, in the context of the pandemic and in response to the Black Lives Matter 2020 campaign, from the perspective of local employment experts of racialized newcomers' employment.

## 4. Intersectionality and Critical Race Theory

We adopted intersectionality and critical race theories to critically comprehend how racism and migration status have significantly impacted racialized communities, particularly in terms of employment, retention, promotion, and ongoing racialized inequities. The theoretical framework of intersectionality originated in the work of African American feminist scholars (Crenshaw 1989; Hill Collins 2000). It provides an understanding of moving beyond single or traditionally preferred categories of analysis, such as sex, gender, race, and class, to include the interactions between different aspects of social identity, as well as the impact of systems and processes of oppression and domination (Hankivsky and Cormier 2009, p. 3). Intersectionality further emphasises that discrimination or oppression does not equally affect people within a particular racial or gender category. Instead, discrimination is understood as an interaction of hierarchies based on various statuses, such as sex, race, class, sexuality, gender, immigration, and disability (Hill Collins 2000; Crenshaw 1989; Hooks 1981; Mohanty 1988).

Critical race theory (CRT) came into prominence in the 1970s from cross-disciplinary scholars who collectively acknowledged how the achievements from the Civil Rights Era were undermined (Delgado and Stefancic 2000). Delgado and Stefancic (2000) highlighted foundational contributing scholars such as Derrick Bell, Patricia Williams, and Neil Gotanda among others. These scholars identified structural aspects of society, explaining the presence and impact of racism and resulting racial hierarchies as driving forces of inequities (Delgado and Stefancic 2000). CRT conceptualises and acknowledges the more subtle nature of racial inequalities within socio-economic hierarchies, historical conditions, individual or group and feelings (Alyward 1999; Delgado and Stefancic 2001; Solomos and Back 2000). CRT allows us to examine how government policies, employers, communities, and even employment agencies might subtly reinforce racial inequalities. Such an approach is important in the Canadian context because the narrative that Canada is a welcoming, multicultural paradise is a longstanding myth that functions to perpetuate various forms of interpersonal and institutional racism by ignoring Canada's problem with racism (Ng and Lam 2020).

Intersectionality is a crucial aspect of CRT and is vital to understanding race inequity (Gillborn 2015). We adopted CRT, in order to study the impact of racism as the key intersection of oppression, as it compounds racial advantages and disadvantages in tandem with other systemic bias. Our goal with this approach is to emphasise the "complex, subtle, and flexible" manifestations and understandings of race and racism (Gillborn 2015, p. 278). Delgado (2011) points out that identity categories could be easily divided, so the uncritical use of intersectionality could lead to the paralysis of critical work amid a mosaic of never-ending differences. To understand how racism works, we need to appreciate how race intersects with other axes of oppression at different times and in different contexts. Therefore, it is critical to use intersectionality as a tool of critical race theory since it ensures we do not overlook one category of oppression when examining others. In addition to racism, this study also explores exclusion based on religious identity and/or socio-

economic status of racialized Canadians. To explore this further, we draw upon empirical data gathered in our study with service providers and employment program coordinators working in the fields of settlement, employment, and equity in Edmonton to understand the barriers experienced by racialized newcomers seeking meaningful employment and the programs and practices in place at the municipal level.

## 5. Employment Support for Racialized Immigrants—Edmonton Case Study

### 5.1. Research Design

This project was initially created as a community-engaged research project with community social workers at the City of Edmonton. However, due to the heightened demands of serving their community during a time of crisis, our community partner members took a step back from driving this research. Our study continued to align with larger lines of inquiry important to our community partners' work in the community and for the City of Edmonton.

### 5.2. Research Questions and Procedure

The data used in this paper coalesced around five interview questions: (1) Do you believe that racialized immigrants face more barriers? If so, what challenges do they face compared to non-racialized newcomers? (2) What strategies does your agency use to help racialized job seekers? (3) How did the COVID-19 pandemic affect job outcomes of racialized immigrants? (4) What do you think about the impact of the Black Lives Matter movement? (5) What policies and strategies could increase the success rate in their work of your programs in helping the integration of racialized Canadians in the labour market?

Our team used non-probability sampling which is a suitable sampling technique for researchers interested in understanding overall patterns across a (cultural) group, for critical case studies, and when "capturing what is considered expert knowledge" (Kirner and Mills 2020, p. 54). The typical sample sizes for 'cultural domain studies' (studies where the focus is narrower as there are fewer 'experts' to engage with) is 10-20 knowledgeable persons (Kirner and Mills 2020, p. 54). Specifically, we used a purposive sampling technique to identify potential participants. Purposive sampling is defined as "sampling that includes all willing participants, so long as they meet the purpose of our sample" (Kirner and Mills 2020, p. 56). Because we were looking for individuals with deep knowledge of the 'behind-the-scenes processes' affecting the settlement and integration of racialized newcomers in Edmonton, purposive sampling is an appropriate technique for this study.

Using semi-structured interviews to collect and analyse non-numerical data (Fossey et al. 2002), our team interviewed nine service providers and employment program coordinators working in the fields of settlement, employment, and equity. Through this research, we sought to better understand barriers facing racialized newcomers seeking meaningful employment in Edmonton, and what programs and practices were in place at the municipal level. We selected municipal-level programs and practices as a case study to address the lack of recent information available about newcomers' experiences of employment before and during the pandemic.

We used grounded theory to analyse the interview responses. Grounded theory is a form of inductive coding in which researchers code important pieces of data (open coding), that are then "piled" into larger categories (axial coding) that make sense in relation to one's data set and the existing literature (Kirner and Mills 2020, p. 132). After we manually coded the data, the research team "analysed the relationships between the codes and identified [key] themes" (Kirner and Mills 2020, p. 132). In our final stage of analysis, our team explored the most significant patterns (key themes) in our data (Kirner and Mills 2020, pp. 140–41). This approach to content analysis helped our team transform "coded qualitative data" into "quantitatively articulated patterns" (Kirner and Mills 2020, p. 132). Grounded theory allows researchers to develop theoretical insights from the data, rather than pre-selecting theories to apply to the data set (Kirner and Mills 2020, p. 132). Grounded theory is particularly useful when analysing data that explores "shared cultural meanings,

norms, and collective practices" (Kirner and Mills 2020, p. 132). In sum, this approach allows us to centre the voices of our participants and follow their lead when identifying important constructs in their experiences.

This approach allows us to explore the role of employment and equity services in facilitating the job searching of racialized immigrants and the hiring process of organisations in Edmonton. We conducted these interviews virtually following provincial and federal health guidelines during the pandemic. We transcribed the recorded interviews. Participants were asked a series of questions around the difficulties they face on a day-to-day basis around challenges in finding and retaining employment.

*5.3. Participants*

Our research team selected and invited (by email) nine participants from employment-service agencies and research institutions in Edmonton, Alberta. All individuals contacted accepted our invitation. We invited individuals from institutions that have offered services to immigrants for at least 15 years. Hence, our participants understood immigrants' struggles and responded to our questions based on their work experiences. They support immigrants and job seekers with different aspects of employment: language instruction, mentorship, networking, job interview preparation, and writing résumés. They also offer information and support with other aspects of their integration and wellness, such as helping clients find housing, childcare, and schools. The success rate of programs is higher than 70%, on average. In general, they measure success based on the number of clients who finish their training and those who get jobs after different parts of the training.[2] The service providers told us that clients working with them usually get entry-level jobs within six months. However, it takes between one and three years for a professional to get a job. Services are provided in French or English.

We identified participants working at either universal, mixed, or targeted service centres. We define targeted service providers as those who provide programming for a specific group of individuals: in this case, services targeted toward newcomers and immigrants. We define universal service providers as those who provide services for anyone, without a specific group identity or category (for example, youth and adult newcomers or Canadian-born residents). We define mixed service providers as those who provide a mix of both targeted and universal services or programs (Esses et al. 2010). (See Table 1).

**Table 1.** Demographic characteristics of participants.

| Participant's Pseudonym [1] | Type of Organization—Service Provided | Racial Identity | Immigration Status |
|---|---|---|---|
| John | Employment Center | Non-White (or Visible minority) | Immigrant—Citizen |
| Michelle | Employment Center | White (or Non-visible minority) | Immigrant—Citizen |
| Rachelle | Employment Center | Non-White (or Visible minority) | Immigrant—Citizen |
| Sarah | Research and Training Institution | Non-White (or Visible minority) | Immigrant—Citizen |
| Jane | Employment, Well-being and Settlement Programs | White (or Non-visible minority) | Canadian born |
| Mary | Employment, Well-being, and Settlement Programs | White (or Non-visible minority) | Canadian born |
| Amanda | Employment, Well-being and Settlement Programs | White (or Non-visible minority) | Canadian born |
| Peter | Employment, Well-being and Settlement Programs | Non-White (or Visible minority) | Immigrant—Citizen |
| David | Employment, Well-being and Settlement Programs | White (or Non-visible minority) | Canadian born |

[1] The research team chose the pseudonyms given to participants.

In what follows, we explore each of these topics in turn: barriers, strategies, the impact of the pandemic, and the potential impact that Black Lives Matter—a campaign that received disproportionate (and, of course, welcomed) attention by the majority community in Edmonton in 2020—had on the awareness around and support for equity-serving work to

address systemic racism in employment practices. Finally, we look into potential practices that might lead to future success in a post-pandemic world.

## 6. Results

### 6.1. Barriers facing Racialized Immigrants or Racialized Canadians

Although all respondents acknowledged that Edmontonian newcomers face barriers when searching for employment, around 50% recognized that racialized job seekers face particular discrimination in the hiring process or are more likely to be underemployed. For example, when we asked if racialized newcomers faced discrimination in the labour market, Jane—one of the respondents, who worked at a targeted service provider, stated, "Yes, they do—100 percent. If I could pick a number higher than 100, I would". Another respondent—John, also working at a service provider with services for newcomers, confirmed this response, stating that "the darker you are, the more discrimination you face....White immigrants are more easily integrated into the majority community—differently from Black communities". Jane confirms this practice of colourism (where the lighter someone's skin colour is, the more acceptable they are for the Canadian workplace):

> "If you're a white immigrant, [you] "just feel more Canadian" because [of] the colour of their skin.... But the demographic of newcomers has changed. Racialized newcomers blend into "our cultural norms" because that's who we are in Canada, but they definitely stand out from the European immigrants that were from many years ago and where my parents immigrated from".

These respondents showcase two important insights: first, that Canadian identity is associated with a white racial identity, which aligns with Ari's (2020) research that mainstream culture in Canada was described as a white, European (ancestry), English-speaking, Christian, capitalist identity with individualist orientations. Second, these excerpts hint at the role (perhaps acceptance) of racialized difference as part of Canada's imaginings of multiculturalism. Srivastava (2007), however, argues that Canadian commitments to diversity are only superficial in what they call the "3D" approach: dance, dress, and dining. Such an approach fails to achieve meaningful engagement with visible differences or equitable access to resources for diverse groups in Canada.

According to our respondents, the most common barriers facing racialized newcomers in the workplace were racism and microaggressions. Peter a respondent working at a mixed service centre, stated that racialized newcomers "want to work but the environment is not designed for them in my mind". Our research respondents specifically identified "foreign sounding" names on résumés and lack of credential recognition as typical microaggressions barring racialized newcomers from entering the job market. He stated, "Regardless of whether you're educated here or not—racialized folks do not get hired at the same rate". These respondents confirmed what research has shown, that racialized newcomers experience exclusion entering the job market based on stereotypes of language fluency and skill aptitude based on perceptions of racialized names and educational context (Oreopoulos 2011; Block and Galabuzi 2018; Ertorer et al. 2020). A lack of Canadian experience was also noted as a barrier for employment, yet, as highlighted by a director at a local research centre (Sarah), "they don't put this requirement [Canadian experience] in job ads anymore, but they do ask about it in the interview itself". This insight—that those on the hiring committee recognize that asking for Canadian experience could exclude certain applicants yet ask for such information "behind closed doors"—signals the continued importance of such a requirement for employers, despite knowing it to be an exclusionary hiring practice. We pick up this thread around anti-racism workplace practices and intercultural training below.

Although almost all respondents noted that a general barrier for newcomers was that they lacked knowledge of cultural norms for Canadian workplaces, some respondents identified "knowing how to respond to microaggressions" as a norm specific to racialized newcomers. Sarah identified white coworkers as the typical perpetrator of microaggressions in the workplace. She gave examples of microaggressions that racialized workers face: "I

remember specifically, this is what I heard from people—engineers saying that during the interview they got questions like "oh so you're from that part of, or so you're Muslim? Are you hard core Muslim? I'm just curious" or "oh your English is good you know?" or "so do you eat curry everyday?"". She stated, "white folks think that they have the right to ask those questions or make those comments". Peter—a service provider at a mixed centre—thought that these microaggressions were worse if the management team had a majority of white managers, regardless of "the existence of intercultural training". It is important to note that the two respondents who identified white colleagues as enacting microaggressions in the workplace were both racialized.

The point above was contradicted by the comments by white respondents. For example, Michelle stated:

> "There are no clients experiencing discrimination because you [have] old people who are very well trained in terms of diversity inclusion. Now when they start working that is a different story....In many cases the discrimination doesn't come from mainstream Canadians, it usually comes from other immigrant groups, which to me was also very surprising cause if you were an immigrant who came years ago, you should be more supportive to new immigrants". (Michelle, targeted service provider)

Using an intersectional lens to explore this data, these excerpts highlight the compounding nature of exclusion for anyone who is both racialized and Muslim (noted from Sarah and Peter comments). Michelle argues that racialized newcomers experience discrimination most often from more established immigrants. This highlights the potential for lateral violence, where discriminatory beliefs about citizenship may overlap with racial discrimination. Michelle's identification of 'old people' was understood to be a reference to those with a history of taking anti-racism education and enacting anti-racism ("diversity inclusion") workplace practices, rather than a statement revealing an intersection of racial identity and age.

Despite acknowledging that racialized newcomers experience different barriers than non-racialized immigrants, few service providers collected data on whether their clients were racialized or provided services specifically for racialized newcomers. For example, Rachelle, who works at a targeted centre, stated, "When we provide our services, we don't keep track records of people's racial profiles, but just by virtue of the clients that we serve, a lot of them are racially profiled". In its 2020 report *Proposed Standards for Race-Based and Indigenous Identity Data Collection and Health Reporting in Canada*, the Canadian Institute for Health Information has argued that organisations which are interested in "monitoring and addressing inequalities that may stem from racism and bias can consider collecting race-based data" (Canadian Institute for Health Information 2020, p. 8). Advocates and scholars have argued that Canada is behind other nations—for example, the United States—in normalising and implementing the collection of race-based data (Owusu-Bempah and Bernard 2021). This follows a report by Edmonton's Social Planning Council, *Confronting Racism with Data: Why Canada Needs Disaggregated Race-Based Data*, which calls for a "coordinated, collaborative framework to collect race-based data as a tool to dismantle racist systems and discriminatory policies within our country" (Edmonton Social Planning Council 2021, p. 2). Barriers facing racialized newcomers in finding gainful employment continue to be the challenges of microaggressions by coworkers, the normalisation of whiteness in Canadian workplaces, and on the side of settlement and integration providers, a lack of information exists regarding the state and need for services for racialized newcomers.

*6.2. Strategies to Address Barriers Facing Racialized Job Seekers to Labour Market Integration*

Institutional and structural rules, policies, and practices have led to systemic racism that deny immigrants and racialized individuals equitable job opportunities (Mooten 2021; Raihan et al. 2023). Despite high expectations of job opportunities upon entering Canada, Vang and Chang (2019) found that immigrants experience racial and ethnic discrimination, an experience which builds the longer they are in Canada. Recent research has found that

income inequality extends to second and third generations, arguing that there are more than just acculturation issues at play (Block et al. 2019, p. 5; see also Ari 2020).

Even with such evidence of the racism facing racialized newcomers, only four institutions had specific programs or strategies to help racialized workers find employment. One such strategy included having open conversations with employers about hiring racialized job seekers. According to Jane, a clear and educative conversation is helpful: "They're people who are just uneducated and aren't aware, and when you provide information and clarity and support that, they become someone different". One institution had an educational program that provided anti-racism training to companies and supported their efforts to design programs tailored to the company's specific context. Another institution focused on teaching job seekers about Canadian workplace culture. This program targeted youth because they were thought to be less likely to receive a job interview. There is also an institution that supports newcomers adjusting to the Canadian workforce culture by following up with their clients for at least three to four years. Mary, a non-racialized respondent, explains how this strategy helps newcomers, particularly those with perceived greater cultural differences, with job retention and adjustment to the Canadian job market. Other institutions focused on guiding racialized newcomers by sharing information about various settlement services available and mentoring them. However, this process can be time consuming and inefficient if there are no specific programs focusing on the particular needs of racialized people. Typical programming include initiating networking events, hosting mentoring programs, and providing language lessons to those who do not speak fluent English. As we will discuss in more detail in the next section, the COVID-19 pandemic drastically decreased the number of events organised by all the institutions.

As a result, advancement in the workplace process of inclusion, diversity, and equity is slow. Organisations with programs that help racialized immigrants find employment struggle, because employers fail to recognize biases in the hiring process. Our participant Jane gave an example of a recent open discrimination case where an employer trying to fill some company positions said, "I would love to talk to you guys [the employment service agency] about how you could help me fill roles that I have vacant, but do not send me anyone that prays". Jane's anecdote refers to job seekers who may need time throughout the workday to meet their religious obligations—a practice most often associated with observant Muslims here in Canada. This statement implies that job seekers can be discriminated against based on their (perceived) religious and racial identity.

Naidu et al. (2022) have argued that "racism is a social determinant of health for newcomers during the acculturation process" (p. 2). Since racism continues to persist in Canadian workplaces, it makes sense that settlement agencies help newcomers understand their rights as employees and new citizens (Kaushik and Drolet 2018), and that employers work to further train their existing employees regarding the existence of white supremacy culture in Canadian workplaces (Canadian Centre for Diversity and Inclusion 2018, pp. 21–22). Such training would need to engage an intersectional approach to account for the interrelation of gender, religion, and racialization when addressing employment marginalisation (Liu 2019).

### 6.3. Impact of COVID-19 on Job Outcomes of Racialized Immigrants

Research has found that the COVID-19 pandemic compounded existing inequalities in the Canadian labour market. For example, Hou et al. (2020) found that racialized individuals suffered greater financial insecurity (compared to the White population) because they were more likely to have precarious and lower-paying jobs, experience job loss or decreased hours of employment (see also, Mo et al. 2020). Lamb et al. (2022) found that recent immigrants were more likely than Canadian-born workers to be unemployed, irrespective of the pandemic (p. 71). Despite this, the authors also conclude that the pandemic did "not dramatically worsen the relative employment or earnings position of recent immigrants" (Lamb et al. 2022, p. 72). The researchers advocate that the labour market disadvantages (pre-pandemic) were already "quite dramatic" (p. 72).

Some employment agencies recognize that the challenge of filling out roles is making employers reconsider their hiring process. For example, with the reopening of the economy after the lockdown there is a high demand for entry-level workers. As such, employers were considering diversifying their employees and learning how to be more inclusive. Participants recognized that discrimination decreasesanytime there is a shortage of workers. David, a mixed service worker, said, "We are getting a lot of employers reaching out to us because they have jobs to fill". John commented that "in 2007 to 2008, there was less discrimination because there were plenty of jobs. Racialized immigrants might get more job opportunities this time around".

The interviewees also recognized that the COVID-19 pandemic had double adverse effects on racialized immigrants. They were the first ones to lose jobs at the beginning of the pandemic, but they were also highly affected by the pandemic as they tend to have essential job positions in hospitals and supermarkets, with a high exposure rate. During the lockdown period, agencies slowed down their activities. Hence, there was less help available to job seekers. Jane stated, "With COVID restrictions, most employees were required to work remotely. Thus, it affected how we delivered our services, the kind of support employment services could provide to job seekers, and the effectiveness of our programs". Hybrid and remote work created new opportunities for racialized workers with required skills who were not getting jobs because employers were not comfortable with their interactions with clients. Peter mentioned that "employers will be like okay, you're Black, you can work from home, you do not have to interact with my client, I am okay with that". According to Peter, virtual work opportunities may have temporarily assuaged anti-Black racism in hiring practices for some clients; however, such practices do not address the systematic anti-Black racism in non-crisis (pandemic) times. Further, such "opportunities" would not assuage the bias of any racialized candidates who did not have sufficient technological skills or access to technology, highlighting the potential intersection of anti-Black racism and socioeconomic status.

Sarah, the director of a local research centre, also confirmed the ambiguous outcome of COVID-19 on racialized newcomers: "Racialized individuals were the first ones laid off during COVID. But at least COVID allowed us to talk about the "r" word [racism]. We are [now] talking about different types of racism, different dimensions". Indeed, recent and racialized immigrants experienced racism and discrimination (Statistics Canada 2020). We believe the discussion around racism increased due to the focus on the news (to follow pandemic updates) and the increased attention by non-racialized Canadians of the murder of George Floyd in 2020 (see below).

These findings suggest that recent, racialized newcomers experience similar if not increased discrimination that compounds with existing, systematic racism in Canadian workplaces (see also, Nangia and Arora 2021). These findings suggest that racism at the intersection of biases associated with newly immigrated individuals—including accent, citizenship, or immigration status bias—feature more prominently in times of (economic) crisis.

### 6.4. Impact of Black Lives Matter (BLM) Movement on Workplace Anti-Racism Practice

The experience of racism in Canadian workplaces depends on how many years the racialized immigrants have lived in Canada. Skuterud (2010) explained that the second generation of visible minority immigrants tend to earn more than the first generation. The Canadian Centre for Delgado and Stefancic (2001, p. 13) found that workplace experiences and discrimination against Black Canadians who have been in Canada for a long period differ from newcomers' experiences. An intersectional analysis is critical to understanding how anti-Black racism and xenophobia compound to affect the racialization of employment practices in any one Canadian workplace (see Liu 2019, for example). The BLM 2020 campaign ignited calls for action regarding long-standing anti-Black racism and presented an opportunity for Canadian workplaces and employers to address in-house systemic racism meaningfully. The BLM movement also influenced the employment of racialized immigrants. Most participants agreed that the BLM movement opened doors to conversa-

tions around racism. It motivated institutions to assess their hiring processes and focus more on setting successful workplace integration for all ethnocultural community members. The participants shared that since the BLM movement began, employment services have received more calls from employers looking for diversity and inclusion training workshops for their companies. For example, Jane, stated that BLM "created awareness, challenged biases within (oneself), encouraged folks to dig deeper, and have honest conversations". Participants also acknowledged that the movement influenced the government's financial support of initiatives related to diversity, equity, and inclusion. The federal government responded by funding community foundations, especially Black-led organisations and programs focusing on Black communities. For example, John, a mixed services provider, stated that "a lot of Black women are going into leadership roles and youth have more access and job opportunities—I hope it continues and isn't just a trend".

The participants recognized that even though there is hope, the system did not change. There is a need for more action and accountability from everyone. Participants admitted that the movement could have been more impactful if it did not coincide with the pandemic. According to many participants, the timing was wrong.

In a review of Canada's official multiculturalism policy, Lei and Guo (2022) argue that multiculturalism has sustained "a racist and unequal society... that tolerates cultural difference but does not challenge an unjust society premised on white supremacy" (p. 1). Mianda (2020) argued that the twin crises of disproportionate racialized policing and loss of Black Canadian (and presumably immigrant) lives due to COVID-19 are "linked expressions of deeply entrenched anti-Black racism" (p. 3). Indeed, anti-Black racism activists and scholars have advocated for the future practice that looks at the intersecting issues of racialization and immigration before, during, and after times of crisis (Bouka and Bouka 2020).

### 6.5. Suggestions for Proposed Policy Meaningfully Address Racism in Canadian Companies

Immigration patterns have made Canada one of the "most ethnically and culturally diverse countries in the Organisation of Economic Co-operation and Development (OECD)", yet the "policy architecture" has paid "little attention to racial economic inequality" (Banting and Thompson 2021, p. 872). Banting and Thompson (2021) argue that universal liberalism (which we understand as taking a colour-blind approach) allows for institutional racism to continue through the "rules, norms and/or patterns of behaviour that perpetuate . . . advantage [for] White populations and disadvantage non-White populations" (p. 876).[3]

In our study, all participants agreed that there is no immediate need for policy change. Based on their experiences, they have noticed that policy change can do more harm than help. For example, policies requiring diversity and inclusion exist in most Canadian companies. However, they are ineffective if hiring teams follow the policies but fail to implement the requirements into their practices and procedures. The interviewees confirm that they always notice the same attitudes, behaviours, and patterns even when employers have policies surrounding equity and inclusion.

> Respondents were asked how the organisation's anti-racism policies have changed over the last 10 to 15 years. Michelle, a working at a mixed service provider, noted, "Now, I'm seeing more statements of accepting applications from diverse populations. Now, whether the policy has actually changed, I have no idea. But there is no [real] change". This point of policy without meaningful change was echoed by the respondent Sarah, the director of the local research centre: "I've seen organisations with policies that were focused on racial equity, and on equity and inclusion in general, but these organisations ended up hiring employees who look like them (white people hiring white people), people who act like them, so the systems perpetuate. There is no room for newcomers, or Indigenous people, no room for racialized folks. If they are hired, they're suffocated by daily microaggressions and microinequities. So, while there are policies, they don't translate into practice.. . .What I am seeing in practice is that you see the same

attitudes, you see the same behaviours, the same patterns even when employers have fancy policies surrounding equity and inclusion".

Despite calls and policies to integrate racially equitable workplaces, our respondents found a lack of meaningful change. Policies are essential, but systemic racism is complex and has many layers. Hence, hiring teams and employees should be trained to introduce a human element to the existing policies. Training should focus on topics related to the value of bringing diverse contributions from a variety of people into workplaces and its impact on the community. Most participants insisted on keeping Canadian companies accountable through audits counting for diversity and workforce inclusion. Advocacy and raising awareness will be helpful to ensure equitable and diverse workplaces everywhere.

Returning to Banting and Thompson (2021), it is important to remember that "overtly racist intentions are not required in order for prevailing structures and institutional arrangements to leave racial hierarchies unchallenged, preserving the many advantages of white Canadians" (p. 886). Indeed, recent research from Southwestern Ontario argues for an evidentiary approach for anti-racism and anti-discrimination initiatives that are contextually dependent (Vaswani et al. 2023; Esses and Hamilton 2021).

### 7. Conclusions

This study investigates the integration process of visible minority Edmontonians into the labour market at a particular moment in time, during and after the COVID-19 pandemic and in response to the BLM 2020 campaign. Questions about discrimination against racialized immigrants in the Canadian labour market have been raised and studied before (Ertorer et al. 2020). However, it is essential to re-examine this problem as it features racism and other intersections of bias, including Islamophobia, and socioeconomic exclusion. This study demonstrated that newcomers to Alberta still face many barriers to employment. In addition, some participants in the interviews recognized that visible minority job seekers face discrimination when searching for employment in Edmonton. Zaami and Madibbo (2021) found similar results when looking at employment barriers facing Black youth in Calgary. These findings demonstrate the existence of intersecting oppresssions of racism, xenophobia, and employment status [similar to Nadeau and Seckin (2010) and Esses et al. (2006)]. They also demonstrate that despite implementing various policies and the recommendations provided by researchers, labour market integration of racialized Canadians has not significantly improved. According to our participant's responses, microaggressions are one of the main challenges most racialized employees continue to face in their workplace. From this research, it is clear that Muslim Canadians face microaggressions in the workplace and that Islamophobia and racism work together to compound bias in hiring. Based on previous studies, Reitz et al. (2022) and Khattab et al. (2019, 2020) found that religious affiliation negatively affects employment. As suggested by our participants, there is still a need for education and training for employers and hiring teams on the importance of inclusion and integration of racialized and Muslim job seekers.

The interviewees confirm the results from past research, such as the Public Health Agency of Canada (2021), which shows that racialized Canadians were more negatively affected by the COVID-19 pandemic. The precarious employment and living conditions of racialized immigrants explain the high level of infection of the COVID-19 virus, death, and unemployment among racialized Canadians. There is an intersection of race, socioeconomic status, and employment status, which explains the negative impact of COVID-19 on racialized Canadians. We also found that the BLM movement was an important event that allowed advocacy of inclusion and integration of racialized immigrants into the labour market and inspired various government agencies to financially support initiatives on equity, diversity, and integration.

Yet, it is essential to hold corporate Canada accountable for their proposed anti-racism commitments following their public support of anti-racism campaigns like Black Lives Matter (Canadian Centre for Diversity and Inclusion 2021; Ng and Lam 2020). Employment agencies explain the struggle of racialized Edmontonians in finding and retaining

jobs by illustrating the unconscious bias of employers, the existence of microaggression, and the lack of recognition of these issues as barriers. While some agencies still do not specifically recognize racialization as a barrier to employment, this study found that some agencies have educational programs providing anti-racism training to hiring teams and employers to normalise workers' cultural differences, while others teach job seekers about the Canadian work culture and keep in touch with newly placed job seekers to facilitate their job integration process.

**Author Contributions:** Conceptualization, D.I., J.L., H.G. and R.D.; methodology, D.I., J.L., H.G. and R.D.; formal analysis, D.I., J.L., H.G. and R.D.; investigation, D.I., J.L., H.G. and R.D.; resources, D.I., J.L., H.G. and R.D.; writing—original draft preparation, D.I., J.L. and H.G.; writing—review and editing, D.I., J.L. and H.G.; visualization, D.I.; project administration, D.I. and J.L.; funding acquisition, D.I., J.L., H.G. and R.D. All authors have read and agreed to the published version of the manuscript.

**Funding:** This research was supported through MacEwan's Strategic Research Grant (2019; RES0000307).

**Institutional Review Board Statement:** The study was conducted in accordance with the Tri-Council Policy on Ethical Conduct for Research Involving Humans, and approved by the Research Ethics Review Board of MACEWAN UNIVERSITY [File No. 101810, approved (June 2020) 10 May 2023].

**Informed Consent Statement:** Informed consent was obtained from all subjects involved in the study.

**Data Availability Statement:** The data presented in this article are unavailable due to privacy and ethical restrictions due to our sample size.

**Conflicts of Interest:** The authors declare no conflict of interest.

## Notes

[1] The concept of "racialized people"is directly related to the definition of visible minority in the Canadian census (Statistics Canada 2023).

[2] Most of the programs run between three and thirteen weeks and teach job seekers the skills they need to land their next employment position.

[3] Colour blind ideology is based on the beliefs of racial neutrality in practices and policies related to economic, social, and political structures (Lee 2022).

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
