# Peer review of "Employment Barriers for Racialized Immigrants: A Review of Economic and Social Integration Support and Gaps in Edmonton, Alberta"

_genealogy, doi:10.3390/genealogy8020040_

Round 1

Reviewer 1 Report

Comments and Suggestions for Authors

I liked this manuscript, and I feel that it has a lot to offer. The writing is overall excellent at the local level (grammar, etc.), though there are some places where word choice or wording could be clearer.  I have typed line-by-line comments below. My overall impression is that the idea of looking specifically at “racialized” immigrants in a specific region/city of Canada is worthwhile, and you provide a good rationale for this focus. As you give evidence, be sure to be clear if the evidence is making (and supporting) a claim about “racialized” immigrants versus immigrants in general. Clarify what some of the terms mean (e.g., “racialized” and how this relates to Indigenous peoples, who are not immigrants but some might argue are still racialized). Finally, in the findings, be sure to give clearer evidence that the participants perceive (or hear from their clients) about microaggressions (see notes below) and that you, indeed, have included an “intersectional” perspective. Make it clearer if you are asking about—and providing—the service providers’ perspectives of their own experience of microaggressions and prejudice or those of the clients with whom they work (or both). Normally, I would break out the writing/grammar comments into their own list in my comments, but there are so few, I will include them all together. There are occasionally places I would use a comma or add a dash, etc. I will not note those sorts of things in this review.

Notably, many of my points are just about the clarity of a phrase here or there. I will note the more important points as I go.

Line-by-line comments

·         10: “racialized.” Not in the abstract, but early in the paper, define what you mean by this term (hopefully with a source). Some authors see “racialization” simply as thinking of people in terms of racial differentiation (e.g., Goldberg). But, by this logic, we could not think of “non-Whites” as “racialized” without also seeing White as a “race.” Thus, Whites would also be “racialized.” This might be a Canadian term; if so, perhaps define it as such.

·         27: “such as…” Some of the examples seem to be “economic classes” and some “programs.” Or is “The Canadian Experience Class” a program? What are “economic class newcomers”?

·         37: “Recent immigration trends” seems to start a new idea. Does this belong in the same paragraph as the previous ideas?

·         39: “As such”—it’s hard to know what this phrase is referring to.

·         41: “Are tasked with”—by whom? (This vagueness is a dilemma with the passive voice).

·         51: “devaluation of qualifications.” I see examples below of discrimination, but I don’t see anything that specifically relates to the “qualifications” of immigrants being devalued. Well, it is there, but it’s more indirect. So, make it clear which evidence demonstrates that the “qualifications” of immigrants are “devalued.”

·         57: Why do visible minorities exclude Indigenous peoples? Justify or explain this choice (unless you are comparing racialized immigrants to domestic Indigenous peoples, but it still seems that some of their issues would be the same).

·         62: “gap in income earnings” –between minority groups? Between minority and non-minority workers? Clarify.

·         65: The logic between the difficulties faced by visible immigrants and the study’s focus on contributions of employment service providers to diversity and inclusion (“For this reason”) is not clear. Also, the findings seem to relate the perspectives of service providers, but I don’t recall seeing their “contributions” outlined in the findings. If you mean the *ideas* that they contribute, perhaps use a different word, as “contribution” is ambiguous.

·         78: “we observed”: In this study? If you are referring to the lit review, I would say “we observed in the literature” or something like that. Often when someone says “we observed” or “found,” I think they are referring to their own primary findings.

·         81ff: Why are the different acts and policies italicized? (I’ll leave this up to you and editors).

·         85: a “fragile narrative” of cultural practices? How are the practices described a “narrative”?

·         96: “racialization and racism”: Define the terms; here, specifically, how is “racialization” alike, different from, or related to racism? (I suspect David Goldberg discusses this, but your own sources probably do as well).

·         101: “consequences for” or “consequences of”—not “consequences on” (you could say “effects on,” perhaps)

·         103-4: “the structural disconnect between policies that claim equity and continues to operate…” Are there some words missing here?

·         112: For several other acts described, the text shows how they were limited in their execution, though the Anti-Racism Secretariat is presented without any disclaimer or outcome. That’s okay with me, but you might also want to treat it consistently as the other legal attempts to ameliorate conditions for the non-White population.

·         By the way, I really liked the deep knowledge that you provide of national and local policies and programs in this paper. This local knowledge is one of the strengths of the paper.

·         132: Maybe word: “…City of Edmonton followed the lead of this province (and all other municipalities…”

·         140-141: What are “landed immigrants”?

·         173: What do you mean by “seniority occupations”?

·         186-187: The transition is clunky: “We covered this. Now we’re going to cover that.” Is there a more conceptual way to get to the next section. Also, the heading for the Lit Review (#3) only appears on line 189, on p. 5 of 13 pages of text, with the rev of lit only being 2 pp. long. A major consideration is to consider shortening intro and making more of what is in the first few pages actually part of the review of literature. At a minimum, rework headings so that intersectionality and CRT is a subheading of Literature Review.

·         198: Most significant issues for immigrants: Yes, but do the study findings apply to “racialized” immigrants, specifically?

·         201: You probably don’t need “refer to.” Also, I would say (202) “minority immigrants”

·         213: any accent, like a French or British accent?

·         225: “more hardship in the labour market”—than whom—racialized men? Canadian Whites?

·         249: “Intersectionality theoretical framework”: Unusual wording. Maybe “the theoretical framework of intersectionality.” A major suggestion is to clarify the relationship between CRT and intersectionality. In fact, here and below, I recommend starting with CRT. Although Crenshaw is one of the main writers who began the discussion of intersectionality (nicely cited alongside Patricia Hill Collins in the manuscript), Kimberlé Crenshaw was also one of the major writers of CRT. Does intersectionality come out of CRT?

·         264-265: “and changing these truths through action was significant.” I’m not quite sure what you mean here by “significant”

·         273-275: It seems like intersectionality is defined twice.

METHOD (Edmonton Case Study)

·          280: “Using semi-structured interviews.” As I understand this type of interviewing, one asks the questions different ways, in different orders, with probes, in more of a guided chat than a set of questions and answers. Yet the MS then lists five specific questions that seemed to be asked of all participants, with the findings structured around these questions. Clarify whether the interviews were, in fact, semi-structured or standardized.

·         282: “our question sought to…” For me, “seeking” requires some form of volition or intelligence. Consider “Through this research, we sought…” or something like that.

·         303-304: Probably semi-colons between main list items, since one contains commas.

·         315-316: “They” and “their’ seem to have different antecedent nouns.

·         313ff: There is a series of statements about the different organizations. I’m assuming that all of these statements apply to all of the programs? Consider building the footnote text into the body of the paper to eliminate the two footnotes.

·         323: Important point: Justify the use of service providers for knowledge (e.g., instead of asking immigrant travelers themselves). Part of the explanation here might get at whether you are seeking to understand the prejudice and microaggressions experienced by OR reported to those service workers.

·         335: Here and in other places, there are parts of the quotation that imply a question. In such cases, provide the context of the question, as you do in another place. “In response to a question about. . . , L. responded, “Yes, they do—.” Also, the letters for the participants reads a little awkwardly (esp as some have two letters and some one). Consider, instead just giving pseudonyms to the participants (though this is a choice of your own writing style).

·         332: “About 50%”: My own philosophy in writing up qual data like this is avoid what Berelson calls “naïve quantification”—giving numerical values to the findings (most, few, etc.) that naïve readers might take as generalizable claims from a sample so small that generalizations cannot be made (nor is that the purpose of this type of research).

·         336 ff: This is a major point: There is reference to “discrimination” and “not integrating.” What you say clearly supports the point about Canadian identity as White (for discussion, you might compare to old studies on British identity, such as There Ain’t No Black in the Union Jack), but there is no clear example of any microaggressions, though line 339 specifically refers to them. If possible, give concrete examples of microagressions and more overt forms of discrimination. There is an example close to the reference to “microaggressions” about “not being hired at the same rate” (362), but this seems more like overt discrimination than a microaggression. Finally, there is a statement about hiring those with “Canadian” experience, which could be a form of microaggression (maybe the term needs a clearer definition in the rev of lit—perhaps using Derald Wing Sue?)—but this sort of policy would also not admit White non-Canadians, so it does not, without a clearer argument, support a claim about microaggressions OR racial discrimination. In other words, through this section, give closer attention to the fit between the claims made and the evidence provided.

·         Interestingly, L’s quote that “confirms” the statement about “discrimination” does not confirm the statement, but suggests that the Canadian demographic has changed and implies that such discrimination might be a thing of the past.

·         378: “those questions…those comments”: Which?

·         403ff: The end of the paragraph seems to be moving into policies or strategies, which is the next main point.

·         416: “Despite high expectations…” dangling modifier.

·         436: “In the next question…” It is only here that I realize that the findings are organized by the interview questions. Certainly, this is an option. I usually encourage my students to arrange the main headings in an “argument”—either topics, cause-effect, and to resist organizing question by question. Still, this is more a stylistic choice.

·         453: Main point; White supremacy. Here, I realized that I don’t recall seeing anything in the findings about intersectionality—e.g., specific issues or experiences of those who are women of Color, who are gay or trans immigrants, etc. Thus, the major point in the rev of lit does not play out in the analysis.

·         461-462: List structure

·         479: “Another side”… another side of what? (Perhaps find a better transition).

·         506: Give more evidence or explanation for the generational differences in racism.

·         507: “Intersectional analysis.” Now the essay comes back to intersectionality. I do see the effects on policy and such of the BLM movement (provide more concrete exemplars for some of the claims, if possible), though still not the link to intersectionality. In some cases, be careful of stating as “facts” things that are participant *perceptions* (e.g., “recognized that…” line 527).

·         540: 5.7: Suggestions for Proposed Policy: how is this section different from “strategies” above?

·         545: perhaps unpack the limitations of the “colour-blind” approach, for any readers who have not yet read or thought about this.

Reviewer 2 Report

Comments and Suggestions for Authors

The topic related to irradicating racism in immigrants’ employment in  Western Canada is highly pertinent.

The participant group (9 individuals) is very small. It is understood that the number of participants in the  group of service providers and employment program coordinators is very limited. This problem could be resolved by involving interviews with racialized immigrants and representing their perspective. However, hearing the voices of the establishment is definitely a worthy goal of the research.

The context description is very well done. So is the literature review, which is very detailed. Intersectionality and Critical Race theory are highly appropriate to the topic. The theories are very concisely and yet comprehensibly described.

Lit review ‘ It is worth noting (138)_ … LINC (142)’ – a reference or two would be helpful, as there is a lot of research on LINC benefits.

Lit review 271 Intersectionality and CRT framework .. para – I would like to see a clearer explanation of how CRT and Intersectionality are interwoven together and what can one expect in terms of innovation coming out from this hybridized approach.

 5. Employment support.

I am entirely missing the Research methodology and Research design. Grounded theory is mentioned, but it needs a brief introduction and justification of its use.

Before procedures, there should be research aims, objectives and research questions. Then methodology with more detail.

The procedures – themes extractions are not explained. Were themes extracted manually or was there any software involved? Do the themes come from the Research questions? How were the subthemes identified? By whom? How were they verified?

Participants are only described in terms of their length of service, but demographic data of particpants should also be provided.

The results need to contain not only excerpts, but some frequencies of the subthemes to prove that ‘most’ or ‘all’ participants identified some themes or ‘the most common responses’ were.

We also have a bit of a mix between ‘responses’ and ‘themes’.

“ Impact of COVID” is definitely an interesting topic, but it should be incorporated in the goals and research questions (which are missing entirely), otherwise it ‘hangs out’ in the article. Consequently, in the Lit review there should be a section on COVID-19 impact on anyone’s employment and immigrant employment and racialized immigrants employment.

The same (incorporate into objective, RQs and Lit review) relates to “Black lives matter” section.

Conclusions are a bit too weak. What a reader wants to see is the breakthrough of the current research. Please highlight the most important results and their significance plus the impact (there is a bit about impact, but name all the stakeholders and how they will benefit from this research

Round 2

Reviewer 1 Report

Comments and Suggestions for Authors

I read this manuscript as if I were reading it fresh, without looking first at my previous comments. I still like the MS, though I still think that it can use a little bit of work. Some of this is up to you and the editors; at a surface level, I could say the MS is ready to go with only a couple of edits. There is a deeper level of thought that you might or might not want to accept at this point in the MS.

This manuscript is overall well-done, with a solid base in the literature (overall) and a solid method and analysis. In my discipline, I often expect a discussion section where the researchers link the findings to previous research and theory, but you have been doing that throughout the findings, which is very common, so I think that is fine.

More importantly, if you do revise, I make three recommendations that I may not have noted on my first review (this is why I think that the study might go forward without these changes): 1) move the rev of lit heading up a bit and provide a clearer sense of flow in the main ideas of the rev of lit; 2) rebalance a bit to give more attention to the rationale for impact of COVID and BLM on the experiences of the racialized minorities; and 3) in findings (and perhaps rev of lit) more clearly highlight the intersectional nature of minority experiences.

As before, my comments are separated by writing (very few comments) and substance, noted by line number.

EDITING COMMENTS

·         377: “policies that. . . continue”

·         542-544: “Between 2001 and 2016. . . over the years.” The statement of the exact years makes “over the years” superfluous (even a little awkward)—just cut the last phrase.

·         734-737: There is something unusual about the grammar in this sentence—perhaps the same noun clause being used as the object of a preposition and the subject of a new clause?

·         1288: Every day (2 words when used as an adj + a noun; one word when used as an adjective: “everyday people”)

·         1290-1365: Put quotation marks around the direct quotation.

·         1616: “Respondent R4.” If “R4” means “Respondent 4,” then are you actually saying “Respondent Respondent 4,” kind of like talking about the “Rio Grande River”?

SUBSTANTIVE COMMENTS

·         196: What do you mean by “receiving society”?

·         204-205: “disproportionate evaluation”—do you mean “disproportionate negative evaluations”?

·         205-215: The preview of the current study (“This study primarily discusses. . . in Edmonton” seems to interrupt the flow of the argument. I would move it to the end of the paragraph.

·         369: There is a clear definition of “racialization.” If possible, support the definition with a source. I believe David Theo Goldberg defined racialization simply in terms of racial differentiation, which might or might not imply racist thought or behavior toward the racial other. However, I checked online, and the usage here is in line with common usage of the term, so I can accept the way you use it throughout the article. Still, a source for the definition would be useful.

·         530-537: The section (Figure 1àend of paragraph) seems to interrupt the flow of thought. In fact, the amount of minorities in the workforce might belong better at the top of the argument. Then, here, the ideas can flow directly from underemployment of those with “ethnic”-sounding names to more details on unemployment rates.

·         542: What do you mean by “economic cycle differences”?

·         652: I get to the fifth page and see a heading for “Lit Review” (suggesting that about 25% of the manuscript is the introduction). Consider moving the Lit Review heading to earlier in the paper. 

·         653ff: This is my main suggestion for the manuscript, and it is the one that you may not choose to address. The material that follows the Rev of Lit heading seems to contain a lot of the same material as what occurs before the heading—that is, disadvantages for racial and ethnic minorities (sometimes coded as those with “ethnic-sounding” names, something I’m guessing applies more to those with Islamic-sounding names than, say, Norwegian- or German-sounding names): lack of transferability of professional credentials, earnings, integration into the workforce. One of the strengths of the rev of lit is the dense basis of the study in previous work with careful citation; this creates a challenge in that the ideas seem to just flow from one to another, with a clear sense of structure difficult to follow.

·         842-860: After four full, dense pages of material on economic and job disparities for “racialized” minorities in Canada more broadly and Edmonton/Alberta specifically, the Rev of Lit spends one brief paragraph justifying a focus on the impact of COVID on the experience of these minorities and 7 lines of type discussing the impacts of BLM (with little background on the movement). Consider rebalancing the coverage.

·         862: Intersectionality: It’s great to include this in the paper. The coverage, including the basis of the approach in CRT, seems quite appropriate. It leads to a clear preview of the rest of the study.

·         METHOD: The design seems appropriate and it is clearly and thoroughly described. I was surprised that you used only Kirner and Mills to justify most of the methodological and grounded theory discussions, but I can live with this. The source is recent, and no doubt K and M account for GT approaches of those who went before them (Strauss & Corbin, Charmaz, and so on). The paper provides solid justification for grounded theory.

·         The sample (nine service workers) seems appropriate for the study, and the sample has good diversity to provide breadth of categories (Lincoln & Guba, 1985).

·         RESULTS: I think in previous comments, I probably recommended use of pseudonyms instead of R1, R2, etc. It personalizes the participants more—but that is your choice.

·         1506: “discriminated against based on their religion.” I suspect that, like prejudice toward ethnic-sounding names, this is also “coded” in that only those of certain religions will receive prejudice (Jewish, perhaps, but more likely in this study, Muslim), which might be racialized (kind of like the U.S. rhetoric on “immigrants” and on “foreigners buying up ‘American’ busineses—which people understand to be non-White immigrants and buyers from non-White nations. No one complains about the Dutch or the English owning businesses in the U.S.)

·         1517-1526: Research on COVID: I recommend moving this to the COVID section of the rev of lit.

·         1626: experience of racism depends on years lived in Canada: Explain more what you mean, here.

·         1629: “An intersectional analysis is critical to understanding…” I agree. Interestingly, most of the rev of lit argument is about specific identities (racial/ethnic, religious minorities), but I don’t recall much there on intersectional oppressions (trans Black women in Canada, for example). And most of the findings are reported in terms of individual opinions; I don’t see much in the results that talks about how women of color might have different experiences than men of color (or LGBTQ+ women of color different than straight or cis-gendered women of color). That is, while I agree that an intersectional approach is useful—even critical—I don’t see much in the rev of lit or findings that actually speaks to an intersectional analysis.

Reviewer 2 Report

Comments and Suggestions for Authors

This version is much improved. I think it merits publication.

Author Response

The reviewer stated "This version is much improved. I think it merits publication."

We thank the reviewer for their feedback.